# The Influence of Sperm Activation Methods and Oocyte Collection on the Reproductive Effects of Northern Pike (*Esox lucius*)

**DOI:** 10.3390/ani15010008

**Published:** 2024-12-24

**Authors:** Michał Cydzik, Krystyna Demska-Zakęś, Mirosław Szczepkowski, Bożena Szczepkowska, Beata Sarosiek, Michał Blitek, Aleksy Kowalski, Radosław Kajetan Kowalski

**Affiliations:** 1Department of Aquaculture, National Inland Fisheries Research Institute, 10-719 Olsztyn, Poland; m.cydzik@infish.com.pl; 2Department of Ichthyology and Aquaculture, Faculty of Animal Bioengineering, University of Warmia and Mazury in Olsztyn, 10-719 Olsztyn, Poland; krysiadz@uwm.edu.pl; 3Department of Sturgeon Fish Breeding, National Inland Fisheries Research Institute, 10-719 Olsztyn, Poland; m.szczepkowski@infish.com.pl (M.S.); b.szczepkowska@infish.com.pl (B.S.); 4Department of Gamete and Embryo Biology, Institute of Animal Reproduction and Food Research Polish Academy of Sciences, 10-243 Olsztyn, Poland; b.sarosiek@pan.olsztyn.pl (B.S.); m.blitek@pan.olsztyn.pl (M.B.); 5Faculty of Medicine, Medical University of Bialystok; 15-089, Bialystok, Poland; alealeksy@gmail.com

**Keywords:** pike, eggs, air stripping, sodium chloride, sperm, fertilization

## Abstract

Pike (*Esox lucius* L.) is a species of great economic and ecological importance. It is a mandatory aquatic predator and one of the most important freshwater fish species. It is worth emphasizing that pike is also a valued aquaculture object, hence, the demand for stocking material has been steadily increasing. These facts support the need to optimize the biotechnics of pike reproduction. Although the reproduction of this species seems quite simple, it causes many problems. Therefore, research is constantly being carried out to improve reproduction, short-term gamete storage, and fertilization techniques. The aim of this study was to use different concentrations of sodium chloride and oocyte collection methods to increase the fertilization rate in pike. The obtained results indicated that sodium chloride solutions can significantly improve the spawning efficiency of pike.

## 1. Introduction

The term assisted reproduction covers all breeding activities carried out in order to obtain healthy hatching from sexually mature spawners [1]. In the case of pike (*Esox lucius*), wild spawners are the most common, which can lead to several challenges. The reproductive period of pike in our latitude takes place in early spring (late March and April) and is strictly dependent on the prevailing weather conditions. Unfavorable thermal conditions can affect the rate of development and quality of the gametes obtained [2,3]. Generally, in natural conditions, male pike are ready to reproduce earlier than females. Sometimes, these disparities in maturation are much greater. Therefore, it is necessary to store semen for a short period of time or even cryopreserve it. Another clearly important phenomenon that reduces the effectiveness of reproduction is the poor quality of gametes.

The quality of the eggs determines their ability to fertilize and, as a result, obtain a healthy embryo [4]. In the case of female fish, including pike, a poor quality of eggs is influenced by a number of factors, including stress [5], immaturity of the eggs, loss of fertilization capacity due to overripeness/aging of the eggs [4], bacterial, fungal. and protozoal infections of roe grains [6], and damage to the eggs grains due to improper wiping of the females.

Obtaining gametes from pike spawners does not pose major problems, and in many countries, this process is similar [7,8]. To synchronize spawning, which may translate to its effects, hormonal stimulation can be performed [3,4,5,6,7,8,9,10]. Gametes are usually collected by massaging of the abdominal wall, after the fish are anesthetized in an anesthetic solution. This procedure reduces the spawners’ susceptibility to stress and significantly facilitates manipulation. However, the improper wiping of females may result in increased egg mortality and oocyte damage due to due to early activation caused by water exposure.

A less invasive method of collecting gametes from female pike is the pneumatic method. This method was first described by Wharton for salmonids in the middle of the last century, but it was not widely popularized [11]. The advantage of this method is that it minimizes manual contact with the abdominal integuments of female fish, thereby reducing the risk of excessive stress and infection [12,13,14]. Pneumatic spawning usually takes longer than “manual” wiping, but this translates to the quality of the eggs obtained and, consequently, the spawning effects. In this case, there is no cracking of egg shells. Generally, the pneumatic method involves inserting a sterile needle into the female’s body cavity. During this time, gas (e.g., oxygen, nitrogen, or air) is introduced from the diaphragm pump at a pressure of max. 0.5 bar. This causes the eggs to be expelled out of the ovary and then through the fallopian tubes and urogenital opening to the outside of the body. An important step that may determine the effectiveness of this method is the verification of the degree of maturity of the oocytes. It should be stressed that, due to the small amount of ovarian fluid, it is sometimes necessary to perform manual spawning. However, the ability to perform air stripping in pike is always a good sign of egg maturity.

Changes in weather conditions have a smaller impact on the quality of sperm than oocytes. A poor quality of sperm usually results from contamination with urine, the premature activation of sperm after contact with water or fluid that activates sperm motility, or incorrect selection of the activating fluid. Determining its quality is practically limited only to visual assessments of the quantity and density of sperm and determining sperm motility [15]. The percentage of motile sperm (MOT) is the parameter taken into account first when assessing the quality of milkweed and its suitability for fertilizing eggs [16,17,18]. A positive relationship between MOT and the percentage of egg fertilization has been found to be a well-known phenomenon [19,20,21,22].

Until the end of the 20th century, the main method for measuring sperm motility was the subjective method. Hence, overestimation errors often occurred due to imprecise assessments of motion [23,24]. Over time and with the development of computer technology, more objective and precise methods have been developed, based on computer systems [25,26,27]. Computer-Assisted Semen Analysis Systems, or CASA for short, enables the objective measurement of the percentage of motile sperm, the speed of their movement, the movement trajectory, the tilt of the head, and the work of the tail. Such a detailed assessment using the CASA system allows for assessing the quality of semen, its fertilization capacity, suitability for short-term storage [26,28] and cryopreservation [29,30], and the effectiveness of the fluids used to activate sperm movement.

The activation of sperm in fish is a process that depends on environmental conditions, especially changes in osmotic pressure. This process is crucial for sperm motility and, consequently, for fertilization success. Freshwater fish sperm are usually activated by a sudden change in osmotic pressure when they transition from seminal fluid (high osmotic pressure) to water (low osmotic pressure). In salmonids, a decrease in the potassium ion concentration during mixing with activation fluid or water triggers motility activation. This change causes water influx into the sperm, causing them to swell and activating their motility apparatus. A higher osmotic pressure (or potassium ions in salmonids), differential between seminal fluid and water, leads to better sperm activation, thereby increasing their motility and ability to fertilize oocytes [31,32]. Similar to sperm, the activation of oocytes (egg cells) in fish is a process dependent on environmental conditions, including osmotic pressure. Osmotic pressure plays a crucial role in the activation of fish oocytes, influencing their maturation and fertilization capacity. A higher osmotic pressure, differential between the follicular fluid and water, leads to better oocyte activation, increasing their readiness for fertilization [6].

In hatchery conditions, the obtained eggs are most often fertilized using water to activate the sperm. The sperm of almost all fish species are non-motile in the testes and seminal ducts. After contact with water or the activating solution, the sperm’s motor apparatus is activated, and their metabolic activity increases. In an optimal environment for sperm, they achieve, among others, their maximum speed, which results in the effective fertilization of eggs. In conditions of assisted fish reproduction, activating fluids such as Jahnichen solution (10 mMTris, 20 mMNaCl, 2 mMCaCl 2, pH 8.5) [33], Tsvetkova solution (50 mMTris -HCl, pH 8.0) [34], Billard solution (125 mMNaCl, 20 mMTris, 30 mM glycine, 1 mMCaCl_2_, pH 7.2) [35], Woynarovich solution (68 mMNaCl, 50 mM urea, pH 7.7) [36], Lahnsteiner solution (100 mMNaCl, 10 mMTris, pH 7.0) [37], Hank’s Balanced Solution (HBSS), and Kurokura solution [38] or commercial Actifish^®^ [39] can be used.

The complex composition of these fluids makes it difficult for aquaculture practitioners to prepare and, consequently, use them. Therefore, the search is ongoing for a new, simple activating fluid, which could be successfully used in hatchery conditions. Herein, among others, sodium salt solutions of two different concentrations were used to activate pike sperm motility.

The aim of this study was to evaluate the effectiveness of using a simple sodium chloride solution in combination with different oocyte collection methods—namely, pneumatic and traditional (‘manual’) techniques—on spawning outcomes.

## 2. Materials and Methods

### 2.1. Spawners, Oocyte, and Sperm Collection

Male and female pike were caught in March with trap tools from Lake Dgał Wielki (northeastern Poland) and then transported to the hatchery of the Sturgeon Breeding Plant, National Inland Fisheries Research Institute, Olsztyn, Poland (NIFRI). The spawners were kept until the experiment was carried out in conditions recommended for this species [7,8]. In total, 16 female and 3 male pike were selected for the study (Table 1). Before the experiment, all fish were anesthetized in a Propiscin (NIFRI, Olsztyn, Poland) solution with a concentration of 0.4 mL/L [40].

Oocytes were obtained from female pike using the traditional method, i.e., by massage of the abdominal wall, and by the pneumatic method, using compressed air. In the case of the pneumatic method, the female was placed in a specially constructed silicone bed at an angle of 30–45. Then, near the ventral fin, a 0.8 mm diameter needle was inserted into the body cavity, through which air was blown (pressure 0.5 bar, flow 0.2 L/min). The injected air, the source of which was a diaphragm pump, caused the eggs to be freely released from the ovary and flow out through the urogenital opening into a sterile bowl.

Before collecting gametes, each female was measured and weighed (Table 1). The mass of the obtained eggs was also determined, and the reaction of the ovarian fluid was checked using an Orion 5 Star pH meter equipped with an Orion Ross Ultra electrode (Thermo Scientific, Waltham, MA, USA). The obtained data were used to calculate relative economic fertility (PGSI), defined as the ratio of the egg mass to the female body mass given in % (PGSI = Wi ((g)/W (g) × 100%).

Procedures involving animals were conducted in compliance with authorized guidelines for the use of experimental animals (68668/2020 MZE-18134 and 68763/2020-MZE-18134). In line with the Polish Protection of Animals Used for Scientific or Educational Purposes Act (January 15, 2015; point 1.2, subparagraphs 1) and 5)), this study was exempted from requiring additional ethics approval.

Semen was obtained by gentle massage of the abdominal wall into sterile syringes and then transferred to Eppendorf tubes. Special precautions were taken not to contaminate the milkweed with urine, feces, blood, or mucus. The protected semen was stored at +4 °C until the next stages of the experiment were carried out.

### 2.2. Activation of Sperm Movement

The semen samples were activated with one of the three following solutions: hatching water, 0.4% sodium chloride solution, and 0.8% sodium chloride solution. The percentage of motile sperm and their movement parameters were analyzed using the CASA system [41]. In order to activate the sperm, 0.2–0.4 µL of semen (depending on initial concentration) was added to 50 µL of the tested activating fluid, and then 1 µL of the resulting mixture was placed on a Teflon-coated slide with 12 wells 30 μm in depth and 4 mm in diameter (Tekdon, Inc., Myakka City, FL, USA). After approximately 5 s, the recording of sperm movement was initiated. A Basler a202K camera (Basler, Germany) was used for recording, which was integrated with an Olympus BX51 camera (Iens Plan FL N 20×X/0.5 NH ph1) (Olympus, Japan). The recording speed was 46.6 frames/s.

The CRISMAS program (Image House Ltd., Copenhagen, Denmark) was used to analyze the first 200 frames from each recording. The program parameters were set as in previous studies [42]. Each semen sample was analyzed twice. Both results were arithmetically averaged for each parameter and each semen sample. Using the CASA program, the following was determined:The curvilinear velocity of the sperm (VCL, µm/s);The straight-line velocity of the sperm (VSL, µm/s);The percentage of motile sperm (MOT, %);The average sperm velocity (VAP, µm/s);The traffic linearity (LIN, %);The amplitude of lateral head deflections (ALH, µm/s);The switch change frequency (BCF, Hz);The percentage of sperm with progressive movement (PRG, %).

### 2.3. Fertilization and Incubation of Eggs

Pike oocytes collected using the traditional method (from 8 females) and the pneumatic method (*n* = 8 females) were divided into portions of 100 eggs each. They were fertilized using the “dry” method, with semen obtained from three males (polled semen) in the amount of 200,000 sperm per egg grain. Before fertilization, the sperm concentration was determined using a Bürker chamber (Sigma-Aldrich, St. Louis, MO, USA) (cytometric method). Fertilization was carried out immediately after collecting oocytes in water from the hatchery and in solutions with concentrations of 0.4% NaCl and 0.8% NaCl (each variant in 3 repetitions). About 3 min after fertilization, the egg samples were rinsed three times with water and then placed in incubation pools divided into troughs. The fertilized eggs were incubated for 10 days in water at a temperature of 12 °C. During incubation, the percentage of fertilization, eyelings, and the percentage of hatched larvae were determined.

### 2.4. Statistical Analysis Methods

The obtained results are presented as the arithmetic mean and standard deviation (±SD). Statistically significant differences in the pH values of ovarian fluid and relative economic fertility were verified with the non-parametric Mann–Whitney test. Differences in the percentage of motile sperm and their movement parameters were analyzed using one-way analysis of variance (ANOVA) and the post hoc Tukey test. In order to characterize the differences in the percentages of eyed and hatched larvae, depending on the method of oocyte collection and sperm activation, a two-way analysis of variance was used. Statistical analysis was performed using GraphPad software Prism 6.02 (GrapphPad Prism Software Inc., San Diego, CA, USA) and Statistica 13.1 software (Statsoft, USA).

## 3. Results

### 3.1. The Influence of Sperm Activation Method on Movement Parameters

The percentage of motile sperm (MOT) immediately after activation with hatching water was 55%. The MOT values were much better after using salt solutions. The average percentage of motile sperm was above 65% (Table 2). In the case of the MOT factor, no statistically significant differences were found. Regarding motility duration, the hatchery water also turned out to be the least effective activator in the case of the percentage of sperm with progressive movement (PRG), whose values fluctuated around 10% and showed differences in relation to the saline solutions (Table 2). The most effective solution turned out to be the 0.4% salt solution, while the 0.8% activating solution was slightly less effective, the values of which were greater than 26%.

The most favorable speed values were characteristic of the 0.4% sodium chloride solution, the values of which were significantly higher than those of the other activators. In the case of the values of the curvilinear velocity (VCL) and average sperm velocity (VAP), 179.5 ± 3.5 µm/s and 172.4 ± 1.2 µm/s, respectively, they were at least 42 µm/s higher than the other activating fluids, thus showing statistically significant differences (Table 2). However, in the case of the straight-line sperm velocity (VSL), the difference was not as significant as in the case of VCL and VAP, but the value was still higher by over 30 µm/s, and, as in the case of the previous parameters, differences occurred (Table 2).

The values of the amplitude of lateral deflections of the sperm head (ALH) of all parameters were similar, the highest salt concentration was the least effective (1.1 ± 0.1 µm), and the highest value was observed with the 0.4% salt solution (1.3 ± 0, 1 µm) (Table 2). The most effective motion linearity (LIN) values were observed for both salt solutions and were similar, at almost 71%, and were almost 25% higher than the hatchery water, where differences were found (Table 2). In the case of the tail switch frequency (BCF), the highest initial value was recorded after the activation of sperm with water 10.7 ± 0.1 Hz, where differences were noted (Table 2).

The results demonstrated a dependency between the NaCl concentration and sperm motility duration. The highest NaCl concentration (0.8%) resulted in the longest sperm motility duration, lasting 120 s. The moderate NaCl concentration (0.4%) led to motility lasting 80 s, whereas in the hatchery water, sperm maintained their motility for the shortest duration—40 s (Figure 1).

### 3.2. The Influence of the Method of Obtaining Oocytes on Economic Fertility and the pH of Ovarian Fluid

The average weight of the oocytes obtained from the female pike using the traditional method was 342.00 ± 205.1 g, and was 9.25 g lower than that obtained using the pneumatic method (Table 3).

The average values of relative economic fertility (PGSI) for both methods of egg collection were quite similar and amounted to 13.8% (pneumatic spawning) and 16.5% (traditional spawning), while the pH of the ovarian fluid was 8.28 (pneumatic spawning) and 8.25 (traditional spawning). The statistical analysis performed did not show significant differences between groups in both the PGSI and spawn pH values (*p* > 0.05).

### 3.3. The Influence of the Method of Obtaining Oocytes and Sperm Activation on the Reproductive Effects of Pike

#### 3.3.1. Percentage of Spawning

The average percentage of eyed eggs traditionally obtained was lower than that after using the pneumatic method (Figure 2). However, statistically significant differences (*p* < 0.05) were found only in the groups where water was used to activate sperm. In these groups, regardless of the method used to obtain oocytes, the percentage of egg occlusion was the lowest (Figure 2). In turn, the best results were obtained after using the 0.4% NaCl solution to activate sperm. Generally, the highest average percentage of egg occlusion was found in those groups where oocytes were collected using the pneumatic method (average 93.7%) (Figure 2). The obtained values did not differ statistically significantly from the values obtained after using the concentration of 0.8% NaCl (88.4%) (*p* > 0.05). Similar trends occurred in the case of traditional roe collection. The mean eyelet percentage after using 0.4% NaCl was 74.5%, and was 4.7% higher than the value obtained using 0.8% NaCl (*p* > 0.05). It should be emphasized that, after using the pneumatic method to obtain oocytes, a lower intra-group variability in the obtained data was observed (Figure 2).

#### 3.3.2. Larval Hatching Percentage

The average percentage of larvae hatching from the eggs obtained using the pneumatic method was higher than that after using the traditional method. At the same time, less variability in the obtained data was observed compared to manual spawning (Figure 3). There were statistically significant intergroup differences between the tested methods (*p* < 0.05), regardless of the type of activating fluid used. The obtained percentage of larval hatching indicated that the most effective fluid for activating the sperm motor apparatus was the 0.4% NaCl solution (Figure 3). In groups where the pneumatic method was used to collect oocytes, it was, on average, 89.5%. This value was significantly different from that obtained after using water (80.3%; *p* < 0.05) and similar to that obtained using the highest salt concentration tested (85%; *p* > 0.05). A similar situation occurred in the case of traditional spawning. The average hatching percentage of larvae after using 0.4% NaCl was 71.2% and was 5.6% higher than the value obtained using both water and 0.8% NaCl by 14.6% and 5.6%, respectively (Figure 3).

## 4. Discussion

This study demonstrated that the pneumatic oocyte collection method (pneumatic spawning) results in a higher percentage of fertilized eggs and hatched larvae of northern pike compared to the traditional method (manual spawning). The application of the pneumatic method combined with the fertilization of oocytes in a 0.4% sodium chloride solution proved to be a highly effective strategy for the artificial reproduction of northern pike.

Research on northern pike has shown that the highest percentage of motile sperm was recorded in a solution with an osmolality ranging from 125 to 235 mOsm/kg [43]. In our experiment, water was found to be a less effective sperm activator for pike compared to the tested sodium chloride solutions (Table 1, Figure 2). The use of salt solutions as activating fluids significantly improved sperm motility. These results indicated that most sperm kinetic parameters were significantly higher when using a 0.4% NaCl solution (136 mOsm) compared to a 0.8% NaCl solution (272 mOsm) and hatchery water (10 mOsm). No significant differences were observed for the MOT parameter, while significant differences were noted for the VCL, VSL, and VAP parameters, which were higher at the 0.4% concentration. The only parameter in which the 0.8% NaCl solution showed slightly better results (by 0.2%) compared to the 0.4% NaCl solution and hatchery water was LIN. It is important to note, however, that the composition and quality of hatchery water may vary depending on the location. Cejko [44] found that hatchery water performed better than Billard solution, but less favorably than Woynarowich solution. Among the many sperm quality parameters assessed using the CASA system, the most crucial for determining the effectiveness of assisted spawning are the percentage of motile sperm (MOT) and sperm linear velocity (VCL) [36]. Previous studies on salmonids indicated that higher sperm velocities are correlated with a higher fertilization success rate [45]. The assessment of motility (MOT) and sperm velocity (VCL) should be a standard procedure in breeding work [18]. Additionally, evaluating these parameters plays a critical role in research aimed at developing or optimizing short-term sperm storage methods [46] and cryopreservation [47,48].

Sodium chloride, when compared to the control sample of hatchery water, in which pike spermatozoa remained motile for 40 s, extended both the duration of sperm motility (up to 80 s) and the percentage of motile spermatozoa, but only at a concentration of 0.4%. A higher concentration of 0.8% sodium chloride further prolonged the duration of sperm motility to 120 s; however, it led to a decrease in the number of motile spermatozoa. Studies suggested that saline solutions are even more favorable compared to the Billard and Woynarovich solutions, where the percentage of fertilization and hatching was lower than that with the use of a 0.4% concentration [44], or at a similar level [49]. A similar phenomenon was observed in Khara’s studies on silver carp (*Hypophthalmichthys molitrix*), where the highest tested salt concentration (0.58% NaCl) resulted in the longest sperm motility duration (40 s), with simultaneously the lowest sperm motility percentage [50]. High salt concentrations likely delay sperm activation, which may lead to incomplete activation, an insufficient energy supply, and, consequently, a reduced motility and lower fertilization success. On the other hand, slower but not excessively delayed sperm activation may contribute to a more balanced ATP consumption, allowing spermatozoa to remain motile for longer, thereby increasing fertilization efficiency by improving viability, achieving better synchronization with the oocyte, reducing sperm damage, and leading to more efficient energy utilization [45]. In freshwater fish, spermatozoa remain quiescent in the male reproductive system due to the osmolality and presence of K+ ions in the seminal plasma [32,51,52,53,54,55]. Typically, sperm motility is initiated in an environment with a lower osmolality than seminal plasma, which is characteristic of freshwater fish [54,56,57]. However, in sturgeons and salmonids, the addition of a few mM K^+^ ions to a hypoosmotic environment completely inhibits sperm motility initiation [32,35,53,54,55,56,57,58,59,60]. The activation of spermatozoa in freshwater fish is strongly dependent on the osmotic conditions of the environment. Changes in osmotic pressure affect sperm motility and their fertilization capacity. In hypoosmotic conditions, such as pure water, a significant difference in osmotic pressure leads to rapid sperm activation, but also to a quick loss of cellular contents, including ATP. This may result in a lower initial sperm velocity compared to activation in moderately osmotic saline solutions, where sperm exhibit a higher initial velocity and longer motility due to reduced ATP losses [61]. Under physiological conditions (seawater for marine fish and freshwater for freshwater fish), the duration of sperm motility varies significantly between species, but is generally limited to a short period. In the literature, the maximum average sperm motility duration for pike is about 80 s, for perch is about 60 s, for carp is about 200 s, and for rainbow trout (*Oncorhynchus mykiss*) is 30 s. The importance of osmotic conditions is also crucial in the context of oocyte activation in freshwater fish, as it significantly affects the efficiency of this process and subsequent fertilization. The oocytes of freshwater fish, such as pike and carp (*Cyprinus carpio*), require precisely defined osmotic conditions for optimal activation. Studies on African catfish (*Heterobranchus bidorsalis*) have shown that moderate salt concentrations (approximately 0.4%) provide the best conditions for oocyte activation, allowing them to achieve a high fertilization capacity and proper embryonic development. Lower salt concentrations (0.1–0.3% NaCl) may result in effective activation, but fertilization efficiency may vary depending on additional environmental factors. In contrast, higher salt concentrations (0.8% NaCl) may induce excessive osmotic stress, leading to incomplete activation or oocyte damage, thereby reducing fertilization success [62]. Based on the results of these studies and other data from the literature [61], it can be concluded that, in assisted fish breeding, the use of specialized activating fluids instead of hatchery water is justified to improve sperm motility parameters and achieve better spawning outcomes in pike.

The results of the conducted studies indicate that there were no significant differences in the number of eggs obtained using the traditional and pneumatic methods. Despite the lack of differences in egg pH between the groups, it was observed that eggs obtained by the pneumatic method exhibited a higher fertilization rate. Egg collection in fish through abdominal wall massage is a standard procedure in the assisted reproduction of species that deposit eggs into the body cavity (Salmonidae) or the intraovarian portion of the oviduct (e.g., cyprinids, percids, pikes, and sturgeons). This procedure is usually straightforward, except for in females with a significant body mass. The use of such females in reproduction allows for the collection of a substantial quantity of eggs, often of a good quality. In the case of traditional egg collection from pike females, an improper and overly forceful pressure on the abdominal wall can lead to the rupture of oocytes located in the ovaries, oviducts, or body cavity. Ruptured eggs are entirely unsuitable for fertilization, as the released nutrients, which should meet the physiological needs of the developing embryo, are lost. Additionally, damaged oocytes result in significant energy expenditures associated with their resorption by the female [63]. The collection of damaged, immature, or overripe oocytes can also negatively affect the remaining healthy oocytes. The mixture of amino acids released into the environment may lead to the formation of visible white “clots”, which can block the micropyles of many oocytes [64] or cause sperm agglutination [65,66]. Another adverse effect of damaged eggs is the reduction in the pH of the ovarian or coelomic fluid [63,67], which may negatively impact sperm motility. In this study, the average pH of the ovarian fluid was 8.25, and did not significantly differ from the pH of eggs obtained by the pneumatic method (*p* > 0.05). It should be noted, however, that the oocytes in this study were traditionally collected by a person with extensive professional experience and high competence, which may have influenced the results obtained. This is also supported by the lack of significant differences in the number of eggs collected, expressed as a percentage of the female’s body mass. Statistical analysis did not reveal significant differences in the relative economic fertility of the female pike subjected to traditional or pneumatic egg collection methods (*p* > 0.05). Despite the lack of difference in the quantity of eggs obtained using both methods (pneumatic and manual), the eggs collected by the pneumatic method demonstrated the highest fertilization rate and larval hatching percentage. The fertilization success of the eggs obtained using the pneumatic method was not only higher, but also showed less variation among individuals (Figure 2 and Figure 3). Therefore, the pneumatic method of egg collection could be recommended for hatchery practices in northern pike reproduction.

In our study, a 0.4% NaCl solution proved to be the most effective medium for fertilizing northern pike eggs. Although hatchery water is the most commonly used solution for fertilization in hatcheries [10], some facilities prefer Billard solution [35] or Woynarovich solution [36,68]. However, simple saline solutions, such as 0.4% NaCl, have also been effectively used for sperm activation and improving egg fertilization rates [3,38,49,50,62,69,70]. The selection of an appropriate activation medium for a given fish species is a critical factor in advancing breeding practices [71]. Optimal activation fluids should reflect the functional properties of seminal plasma and ovarian or coelomic fluids, which have the greatest impact on sperm quality and motility. Woynarovich solution is traditionally used for fertilizing the eggs of cyprinids, percids, and pike, and Cejko’s [44] research confirms its effectiveness, showing a higher fertilization rate of pike eggs compared to Billard solution. Similarly, in the case of European perch (*Perca fluviatilis*), Woynarovich solution proved to be the most effective medium for egg insemination [72]. The present studies suggest that saline solutions, particularly 0.4% NaCl, can be successfully used for both sperm activation and the fertilization of pike eggs. Considering its low cost, ease of preparation, and high effectiveness in assisted spawning, this method can be recommended as an effective means of fertilizing pike eggs. The high rate of egg fertilization and larval hatching achieved with the use of a 0.4% NaCl solution confirms its effectiveness, regardless of the method of oocyte collection. Saline solutions, such as 0.4% NaCl, outperform both Billard and Woynarovich solutions, as confirmed by the studies of Łuczyński [49].

The results of the experiment indicate that the pneumatic method may yield better outcomes compared to the traditional method of oocyte collection in northern pike. Following the application of this method, the average percentage of fertilized eggs was higher compared to the manual method, regardless of the sperm motility activator used. Statistically significant differences at this stage of embryogenesis were observed only when water was used as the sperm activator. However, the pneumatic method had a greater impact at the hatching stage, where, regardless of the activation fluid used, the hatching rate was higher than that in traditional spawning, as confirmed by statistically significant differences observed across all groups (*p* < 0.05). The pneumatic method was originally developed for fish whose eggs ovulate into the body cavity, such as salmonids. Its first application took place in the 1960s in Australia, where Wharton used air pressure to collect eggs from rainbow trout [11]. Despite the method’s great potential, its application has not been widely documented in other fish species. Preliminary studies on its effectiveness and potential adaptation in rainbow trout production were conducted by Leitritz and Lewis [64],. In the case of Chinook salmon (*Oncorhynchus tshawytscha*), it was demonstrated that, when using the pneumatic method, the issues of ovarian fluid turbidity and the rate of ruptured eggs are negligible [73]. The pressure method also proved effective in rainbow trout, fish of the genus Salmo (e.g., sea trout (*Salmo trutta morpha trutta*), brown trout (*Salmo trutta morpha fario*)) [74], and whitefish (*Coregonus lavaretus*) [12]. Although the pneumatic method was primarily patented for fish species that ovulate eggs into the body cavity, its application in species with oviducts, such as pike, is also feasible. The studies by Cejko [44] and those presented in this work on pike confirm its effectiveness.

In summary, the pneumatic method of egg collection offers several advantages, particularly for fish with large eggs, minimal ovarian fluid production, and low-viscosity eggs. When performed correctly, this technique enables the rapid collection of oocytes while preserving their high quality. Additionally, it minimizes mechanical damage to the fish, such as abrasions and the removal of mucus, which reduces the risk of infections and post-spawning complications. Despite these advantages, the study results suggest that the egg collection method alone does not determine overall reproductive success. The combination of this method with appropriate sperm activation and fertilization procedures is essential for achieving a high spawning efficiency in pike. It is the synergistic interaction of these factors that leads to optimal reproductive outcomes in pike.

## 5. Conclusions

Based on the research conducted, the following can be concluded:The use of salt solutions enhances pike sperm motility. Higher osmotic pressures (e.g., 0.4% and 0.6% NaCl) improve fertilization by extending sperm movement and ATP efficiency, particularly at 0.4%. However, overly high concentrations (e.g., 0.8%) can slow or hinder activation, reducing motility and fertilization success.A prolonged sperm motility contributes to better fertilization by increasing the sperm lifespan, which especially helps with the fertilization of larger volumes of gametes by increasing the time necessary for mixing the gametes (sperm and eggs)The pneumatic method of obtaining pike oocytes positively impacts spawning outcomes, measured by the percentage of eyed eggs and hatching,The most effective method for the assisted spawning of pike tested in this study involves obtaining eggs pneumatically and fertilizing them using a 0.4% NaCl solution.

## Figures and Tables

**Figure 1 animals-15-00008-f001:**
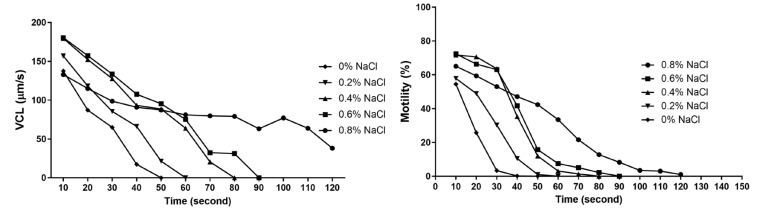
Curvilinear velocity of spermatozoa (VCL) and the percentage of motile spermatozoa (MOT) after exposure to water and various salt solutions as activating fluids (mean values).

**Figure 2 animals-15-00008-f002:**
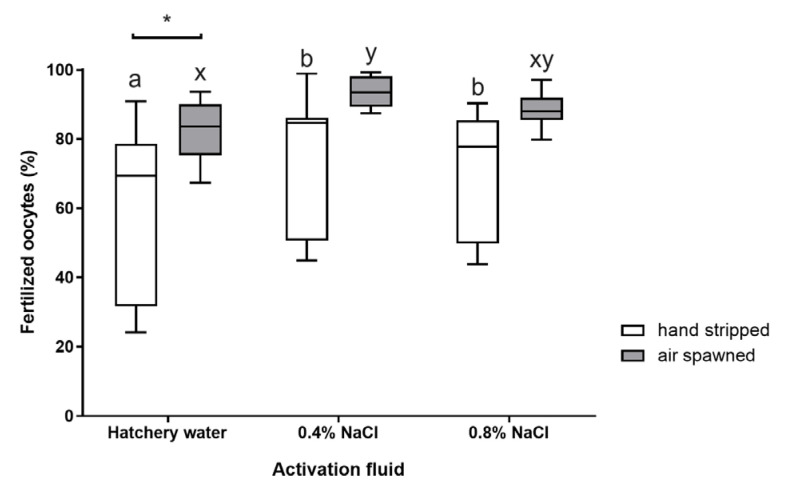
The percentage of eyed pike eggs obtained using the traditional and pneumatic methods. Values marked with the same letter index for a given spawning method do not differ statistically significantly (*p* > 0.05); * significant differences between groups were marked (traditional vs. pneumatic; *p* < 0.05).

**Figure 3 animals-15-00008-f003:**
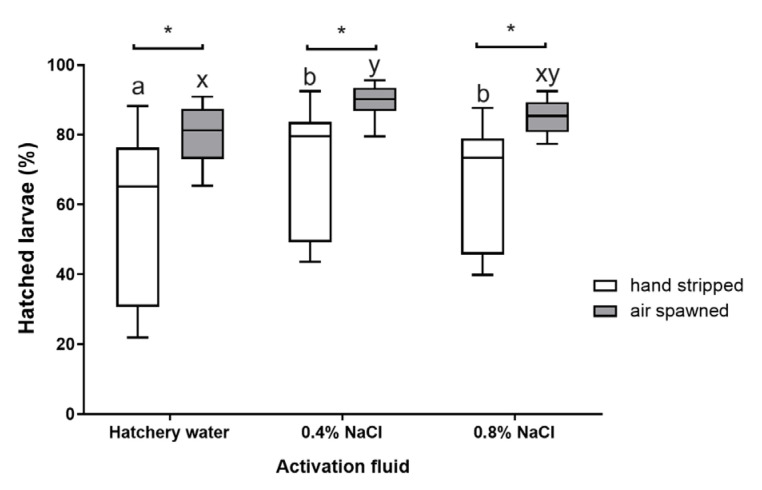
Percentage of hatched pike larvae after using the traditional and pneumatic methods of obtaining oocytes. Values marked with the same letter index for a given spawning method do not differ statistically significantly (*p* > 0.05); * significant differences between groups were marked (traditional vs. pneumatic spawning; *p* < 0.05).

**Table 1 animals-15-00008-t001:** Weight (W) and body length (Lt and Lc) of pike spawners (mean values ± SD).

Sex	n	W (g)	Lt (cm)	Lc (cm)
Female	16	2078.62 ± 806.69	65.56 ± 8.86	56.81 ± 7.60
Male	3	966.67 ± 266.56	54.00 ± 6.56	47.00 ± 5.57

**Table 2 animals-15-00008-t002:** Computer analysis (CASA) parameters of pike semen recorded in 0.8% NaCl solution, 0.4% NaCl solution, and hatchery water (mean values ± SD).

	Activation Fluid		
Parameter CASA	0.8% NaCl	0.4% NaCl	Hatchery Water
MOT (%)	65.1 ± 12.0 ^a^	71.7 ± 11.0 ^a^	54.5 ± 7.3 ^a^
PRG (%)	26.7 ± 10.5 ^b^	30.7 ± 0.1 ^b^	10.0 ± 1.2 ^a^
VCL (µm/s)	132.8 ± 14.5 ^a^	179.5 ± 3.5 ^b^	137.5 ± 16.6 ^a^
VSL (µm/s)	95.5 ± 14.5 ^a^	128.3 ± 6.3 ^b^	77.0 ± 8.3 ^a^
VAP (µm/s)	125.6 ± 15.8 ^a^	172.4 ± 1.2 ^b^	127.0 ± 16.0 ^a^
ALH µm	1.1 ± 0.1 ^a^	1.3 ± 0.1 ^a^	1.2 ± 0.2 ^a^
LIN (%)	70.9 ± 4.5 ^a^	70.7 ± 4.6 ^a^	56.4 ± 1.6 ^b^
BCF (hz)	8.0 ± 0.5 ^a^	8.7 ± 1.1 ^a^	10.7 ± 0.1 ^b^

MOT = sperm motility, PRG = sperm progressive motility, VCL = curvilinear velocity, VSL = straight linear velocity, VAP = average path velocity, ALH = amplitude of lateral head displacement, LIN = movement linearity, and BCF = beat cross frequency. a,b values with different letters indicate significant differences between groups (*p* < 0.05).

**Table 3 animals-15-00008-t003:** Weight (W) and body length (Lt and Lc) of female pike and the mass of collected oocytes (Wi) (mean values ± SD).

Method of Obtaining Oocytes	n	W (g)	Lt (cm)	Lc (cm)	Wi (g)
Hand stripped	8	1956.75 ± 850.48	64.62 ± 10.68	57.12 ± 9.42	342.00 ± 205.10
Air spawned	8	2200.50 ± 762.90	66.50 ± 7.04	56.50 ± 5.80	351.25 ±167.11

## Data Availability

Data will be available upon request from the corresponding author.

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
