# Peer review of "The Influence of Sperm Activation Methods and Oocyte Collection on the Reproductive Effects of Northern Pike (*Esox lucius*)"

_animals, 2024, doi:10.3390/ani15010008_

Round 1

Reviewer 1 Report (Previous Reviewer 3)

Comments and Suggestions for Authors

One of the objectives of the Materials and Methods section is to enable other researchers to replicate the study and validate the results. Therefore, it is essential to provide a thorough description of the experimental design, sampling, procedures, and techniques employed. A detailed account of the materials and methods ensures the study's replicability.

When novel methods and protocols are described, they must be rich in detail. Conversely, well-established methods should be described briefly and appropriately cited.

In the revised version presented, the authors have not adequately addressed the requests of this Reviewer. While this Reviewer understands that some points may be discretionary for the authors, those that compromise the replication of the methodology cannot be overlooked.

Some responses to this Reviewer's comments are not understandable and even contain contradictions.

The term "controlled reproduction" is incorrect, with "assisted reproduction" being the universally accepted and understood term in animal reproduction.

The considerations regarding the methodology have not been clarified in the manuscript, nor appropriately referenced (e.g., pneumatic method, slide for semen analysis, proper validation of CASA as provided references do not validate). This issue is a significant barrier to accepting this manuscript for publication. This methodological flaw results in the impossibility of replicating the results, which is a cornerstone of the scientific method.

Therefore, given the significant gaps in the methodology, this article does not allow for study replication and should not be accepted in its current form for publication in this journal.

Comments on the Quality of English Language

Same as before as no changes was made:

The technical and scientific use of English is adequate. However, there are some areas where the text could be improved for greater clarity and fluency. In particular, the structure of some sentences is complex and could be simplified, and there is repetition of ideas that could be condensed. Additionally, the use of certain expressions could be adjusted to improve readability and the flow of the text. Thus, adjustments are necessary to enhance the clarity and fluency of the text without compromising the technical content presented.

The reviewer recommends that the text be reviewed by a native English speaker with knowledge of technical terms.

Author Response

Comments 1: Some responses to this Reviewer's comments are not understandable and even contain contradictions.

Response: 

Thank you very much for your detailed feedback. We acknowledge your concern regarding potential contradictions in our responses. However, we kindly ask for clarification on the specific points or sections where these contradictions were observed. This would help us better understand your perspective and provide more precise explanations or corrections where necessary.

We greatly appreciate your time and consideration and remain committed to improving the clarity and consistency of our response.

Comment 2. The term "controlled reproduction" is incorrect, with "assisted reproduction" being the universally accepted and understood term in animal reproduction.

Response 2: 

We appreciate your comment on the use of terminology. We understand that the term "controlled reproduction" was not appropriate, and we have revised the text to reflect the correct terminology—“assisted reproduction,” which is universally accepted in the field of animal reproduction.

Comment 3. The considerations regarding the methodology have not been clarified in the manuscript, nor appropriately referenced (e.g., pneumatic method, slide for semen analysis, proper validation of CASA as provided references do not validate). This issue is a significant barrier to accepting this manuscript for publication. This methodological flaw results in the impossibility of replicating the results, which is a cornerstone of the scientific method.
Response 3: 

Thank you very much for your insightful feedback regarding the validation of the CASA system for fish sperm motility analysis and the pneumatic method for inducing spawning. We appreciate the opportunity to clarify our approach and the considerations involved.

We recognize that validation within CASA systems can be valuable for specific diagnostic applications. However, when assessing fish sperm motility, achieving reliable results depends heavily on maintaining consistent system settings. In our research, we utilized the CASA system with parameters precisely as described in the referenced publication (we added new one indicating the specific settings of the programme used). Under these standardized conditions, we believe that any researcher using the same equipment and settings should obtain comparable results.

Regarding the pneumatic method, we consider that specifying key parameters such as gas pressure, injection site, needle diameter, and gas flow rate provides sufficient detail to replicate the procedure accurately. The concept of a formal “validation” process for this method appears difficult to define in this context.

Additionally, the glass slides used in our study are commonly manufactured by TEKDON, and we have provided the relevant specifications and manufacturer details to facilitate replication.

We kindly suggest that the reviewer reconsider the request for additional validation based on these explanations. We remain open to further discussion and clarification if needed.

Comment 4: The reviewer recommends that the text be reviewed by a native English speaker with knowledge of technical terms.

Response 4: 

We have made revisions to enhance readability, including simplifying complex sentences and correcting grammatical issues with the help of a native speaker.

Reviewer 2 Report (Previous Reviewer 2)

Comments and Suggestions for Authors

I have previously reviewed this manuscript, and my suggestions have been incorporated into the revised version. A few comments are given below:

-NaCl solutions have not been compared with other common activating fluids, which could limit our understanding of the optimal activation method. -Long-term effects on embryo development and larval survival were not investigated, which is critical for evaluating the effectiveness of the methods.

Author Response

Comments 1: -NaCl solutions have not been compared with other common activating fluids, which could limit our understanding of the optimal activation method. -Long-term effects on embryo development and larval survival were not investigated, which is critical for evaluating the effectiveness of the methods.

Response: 

Thank you for your valuable suggestion. The primary objective of this study was to evaluate the effectiveness of sodium chloride (NaCl) solutions as sperm-activating fluids for northern pike, particularly focusing on the concentration ranges that optimize sperm motility and fertilization success. The selection of NaCl solutions for this study was motivated by their simplicity, cost-effectiveness, and relevance to hatchery practices. While comparisons with other activating fluids such as Billard and Woynarovich solutions were not within the scope of this research, we recognize that such comparisons could provide additional insights into the optimization of activation protocols. This limitation is now explicitly mentioned in the revised discussion as a potential area for future research.

We appreciate the reviewer highlighting the importance of long-term studies on embryo development and larval survival. While this study primarily focused on immediate outcomes such as fertilization rate and hatching success, we acknowledge that long-term observations are critical for a comprehensive evaluation of reproductive techniques. We have incorporated this limitation into the discussion and identified it as a future research direction. Expanding the scope to include parameters like larval growth, survival, and health under varying environmental conditions will provide a more complete understanding of the methods' effectiveness.

Reviewer 3 Report (Previous Reviewer 1)

Comments and Suggestions for Authors

1、This is an interesting study, and the author may consider adding "graphic abstract" to enhance the readability of the article for readers

2、Line 461-462, 465-466  The format of the reference citation is incorrect. There are also other similar problems.

Comments on the Quality of English Language

The English could be improved to more clearly express the research.

Author Response

Comments 1: This is an interesting study, and the author may consider adding "graphic abstract" to enhance the readability of the article for readers.

Response1: 

Thank you for your suggestion to include a graphic abstract. We agree that a well-designed graphic abstract could enhance the readability and accessibility of the manuscript for a broader audience. While we did not initially include one, we will consider adding a graphic abstract in the final version of the manuscript to provide a clear visual summary of the main findings and methodology.

Comments 2. Line 461-462, 465-466  The format of the reference citation is incorrect. There are also other similar problems.

Response 2: We appreciate your careful attention to the formatting of references. We acknowledge that some references were incorrectly cited in the manuscript, particularly in lines 461-462 and 465-466. We have thoroughly revised the reference list and ensured that all citations are consistent with the journal’s guidelines. These corrections have been made throughout the manuscript to improve clarity and accuracy.

Comments 3. The English could be improved to more clearly express the research.
Response 3. 

Thank you for pointing out that the English language could be improved. We have carefully reviewed the manuscript for clarity and precision in language. We have made revisions to enhance readability, including simplifying complex sentences and correcting grammatical issues with the help of a native speaker.

Round 2

Reviewer 1 Report (Previous Reviewer 3)

Comments and Suggestions for Authors

In this revised version, the authors have provided necessary information regarding the use of the CASA system and have properly referenced previously unreferenced materials and methods (pneumatic method and CASA use). Although this reviewer has some questions about the correct use of the CASA system, it is not within this reviewer's purview to question previously published articles.

There has also been a considerable improvement in the English writing, the correct use of “assisted reproduction”, and this reviewer appreciated the addition of the concluding paragraph in the discussion.

Therefore, this reviewer commends the authors for the improvements made and for providing references to the procedures adopted. By referencing prior studies, even if not described in detail, the authors facilitate the replicability of the experiment.

This manuscript is a resubmission of an earlier submission. The following is a list of the peer review reports and author responses from that submission.

Round 1

Reviewer 1 Report

Comments and Suggestions for Authors

Author Response

A brief summary:

This MS by Cydzik et al. compare the quality of pike roe collected using the traditional method (abdominal integument massage) and the pneumatic method. Moments after collection, the pseudo ganodosomatic index (PGSI) was measured. Additionally, various concentrations of sodium chloride used to activate sperm movement and their impact on oocyte fertilization under controlled reproduction conditions were tested. This study found that sodium chloride solutions significantly improve the motility of pike sperm and have a positive effect on spawning effects measured by the percentage of eyed eggs and hatched larvae. The most effective method of reproducing pike under controlled conditions is to obtain oocytes using the pressure method, while using a 0.4% NaCl solution to activate sperm. Overall, I thought this was a well-executed study in a system with limited previous knowledge of this level on this specific topic. I appreciated their multi-faceted approach and the time course involved in this work and think it add significant merit to their work. I do think that their overall conclusions were related to results. Overall, I think this is good work, but should undergo some revision before acceptance.

1) General concept comments. I think the author should seek help with writing in English, I hope round of editing with English native language would further improve the quality of the writing. The English grammar should be checked again. I also have a couple suggestions.

2) Specific comments Abstract: There are some experimental data and significant differences in the Results part. However, there is no any information here. I suggest the author rewrite “Abstract” and introduce the result information more clearly. Generally, the “Keywords” should try to avoid repetition with the words in the “title”, aiming to make it easier for readers to search for your articles in Google.

  1. Introduction Line 57 “[3]]” change to “[3]” Reference [10] is missing here.

Corrected, we changed “[3]]” to “[3]” and references have been updated.

  1. Materials and methods

Line 136: Numbers cannot be placed directly at the beginning of a sentence. “16 female and 3 male pike” change to “The 16 female and 3 male pike”

Corrected, we changed “16 female and 3 male pike” change to “The 16 female and 3 male pike”

Line 136-138: I suggest add “Reference” to explain the dose of Propiscin. Line 142: “angle of 30-45°” change to “angle of 30-45” or “30-45°” Table 1. “7.6” change to “7.60” The unit format should be unified. Eg.

Corrected, we changed “angle of 30-45°” to “angle of 30-45”, we changed “7.6” change to “7.60” and we added “Reference” to explain the dose of Propiscin.

Line 161 the author use “4°C”, however, they use “12 ° C” in Line 196, completely different formats. I suggest to revise and check this part again.

4 degrees refer to the storage of pike sperm, while 12 degrees is the incubation temperature of fertilized pike eggs.

  1. Results

Line 244-247: “Figure 2” change to “Figure 1” In Figure 1, the data was expressed with only “mean values”, but not “mean values±SD”, why? There is also not statistically significant differences analysis and instruction in Figure 1 and its result. I suggest the author explain this problem clearly.

Corrected, we changed “Figure 2” to “Figure 1”

  1. Conclusions

This is too long, I suggest to reduce and simplify, and only keep important information.

Corrected, we reduced and simplified

 Check the Reference again. Some references such as 12 are too old. 6 Reference Check the Reference again. There are about 10 References in Polish. I suggest to delete or replace some, and add Reference in English.

Checked, corrected and added Reference in English.

Reviewer 2 Report

Comments and Suggestions for Authors

The manuscript "The influence of sperm activation methods and oocyte collection on the reproductive effects of northern pike (Esox lucius)" presents a comprehensive study on the reproductive strategies for northern pike (Esox lucius), focusing on the comparison of traditional and pneumatic methods for oocyte collection and the effects of sodium chloride (NaCl) concentrations on sperm activation. The authors aim to optimize controlled reproduction techniques to enhance fertilization rates and larval hatching success, addressing a significant need in aquaculture due to the increasing demand for northern pike.

Overall, the manuscript presents valuable research that contributes to the field of fish reproduction and aquaculture. The findings are likely to be of interest to researchers and practitioners in aquaculture. For this reason, it is not suitable for animal journal and on the other hand, it does not have enough data and results for this journal.

 Comments

-      While the manuscript references relevant studies, a more extensive review of recent literature on similar reproductive techniques in other fish species could strengthen the introduction and discussion sections. This would provide a broader context for the findings and highlight the novelty of the research.

-      The methodology section could benefit from additional detail regarding the specific conditions under which the experiments were conducted (e.g., water quality parameters, temperature control during sperm activation). This information is crucial for reproducibility.

-      The discussion could delve deeper into the implications of the findings for practical applications in aquaculture. For instance, how might these optimized techniques influence breeding programs or conservation efforts for northern pike?

-      Figures and Tables: providing more visual data representation could help in illustrating key findings.

Author Response

We have revised the manuscript according to the reviewer's comments

Reviewer 3 Report

Comments and Suggestions for Authors

1. Brief Summary

This manuscript investigates the efficacy of different sperm activation and oocyte collection methods on the reproductive outcomes of the pike (Esox lucius). The study demonstrates that the pneumatic method of oocyte collection, combined with a 0.4% NaCl solution, improves fertilization and larval hatching rates. However, the study was conducted nearly a decade ago, and most references are outdated, which may render the manuscript obsolete before publication.

2. General Concept Comments

The manuscript identifies gaps in knowledge regarding sperm activation and oocyte collection in Esox lucius but needs to address the limitations of current methods and consider recent advances, such as commercial media for sperm activation. Why use the NaCl solution (possibly produced rather than a commercial product) when it is well-known that K+ and ions are key factors in controlling spermatozoa motility initiation?

The opening sentence of the introduction caused significant confusion for this reviewer. When the authors state, "The term controlled reproduction covers all breeding activities carried out to obtain healthy hatching from sexually mature spawners", what exactly do they mean by "controlled reproduction"? Does it cover semen collection and ova insemination? This was unclear in the first paragraph of the introduction.

For this reviewer, "controlled reproduction" is understood as the management and control of environmental and physiological conditions to stimulate the natural reproduction of fish, where human intervention is minimal. In contrast, "assisted reproduction" refers to more direct intervention techniques to facilitate reproduction, which may include hormonal induction to stimulate ovulation and spermiation, artificial fertilization, and manipulation of gametes (ova and sperm) outside the fish's body. Therefore, there is significant human intervention.

The term “assisted fish reproduction” is used by the last author, in Kowalski & Cejko, in the article “Sperm quality in fish: Determinants and affecting factors.” Therefore, it is necessary to clarify why the authors believe “controlled reproduction” is the correct term or to change it to “assisted reproduction.”

The manuscript lacks a description of the control of experimental variables, such as temperature and pH. Additionally, the sample size is small, compromising statistical significance. The methodology does not clarify the use of sterile gases or the preparation of saline solutions, raising questions about the sterility and safety of the procedures.

The description of sperm activation solutions in the methodology is vague, and the choice of NaCl concentrations is not justified. Reference [33] does not address semen analysis with the CASA system, and the use of slides for analysis requires more details.

The literature review is inadequate, with only two recent references among 69 (45 of which are over ten years old). In a dynamic field like aquaculture, it is essential to update references to reflect the current state of knowledge. Including more recent studies is necessary to contextualize the results.

Due to incorrect numbering of references, this reviewer could not adequately assess the discussion section, as it was impossible to verify the references supporting the discussions. Therefore, this reviewer is unable to evaluate the Discussion section of the present manuscript.

3. Specific Comments

Abstract: The opening sentence creates confusion about what "controlled reproduction" covers. It is recommended to clarify or change to "assisted reproduction."

Lines 49-54: Use continuous text instead of bullet points.

Lines 55-57: Using these lines as an example, this is "assisted reproduction," not "controlled reproduction."

Lines 69-70: For oxygen or nitrogen, they may be used with medicinal grade—is this the case? How is ambient air sterilized? It seems nonsensical to have a sterile needle if sterile gas isn't used. Could this be clarified? Clarify the use of sterile gases and the sterilization of ambient air. Discuss the choice between oxygen and nitrogen. Is it nitrogen or nitrous oxide? Also, why oxygen or nitrogen? In mammals, CO2 is used as it can be absorbed by tissues and removed through the respiratory system. How does the application of oxygen and nitrogen work?

Line 85: Old reference. Retain but include new ones. Suggestions:
https://doi.org/10.1016/j.theriogenology.2023.12.016
https://doi.org/10.1016/j.anireprosci.2019.01.001
https://doi.org/10.1016/j.anireprosci.2022.107018

Lines 86-88: Not only this. See https://doi.org/10.1016/j.therwi.2024.100091 (interesting considerations about CASA), where it is mentioned that “Evaluation of sperm by conventional microscopy is susceptible to subjectivity, causing significant discrepancies across studies.” If it were only errors of overestimation (and also underestimation), it would be less problematic than the impossibility of replication or replicability of results, which is a cornerstone of the scientific method.

Lines 116-121: Use continuous text instead of bullet points.

Lines 117-121: These formulas are from the last century. The newest is 25 years old! In the last two decades, assisted reproduction has evolved considerably, with low-cost, high-efficiency commercial options available. It is important to address this (commercial stable media for sperm activation, such as ActiFish).

Line 123: Would it be “fish farmer”? This Reviewer understands "fishing practitioners" refers to people who engage in fishing. This can include professional fishermen, amateurs, or anyone involved in the activity of fishing. What is the objective of a fisherman collecting and activating semen? And being a fish farmer, why not use commercial activation media developed specifically for this purpose, which is not expensive?

Line 136: When a number starts a sentence, write the number in full. Also, when referring to two or more animals, use plural (males and females). Correct to: "Sixteen females and three males."

Line 138: The use of capitalization in units like mL, L, and μL is guided by conventions in scientific and technical writing to ensure clarity and avoid confusion. The "L" is capitalized to avoid confusion with the number "1" or the lowercase letter "l," which can look similar in certain fonts. Please correct throughout the manuscript.

Lines 139-145: The description of the pneumatic method of oocyte collection lacks details on how air pressure is controlled and the possible effects on the oocytes.

Line 141: Include an image of the pneumatic method for better understanding.

Line 143: Ambient air? Without an air filter to avoid contaminants? As previously questioned, what is the risk to the animal of insufflation with non-sterile air? What is the risk of using non-sterile air and how is this managed?

Line 146: Use "gametes" instead of "sexual products."

Lines 152-156: This paragraph would be better positioned after “2. Materials and Methods” before the section "2.1. Spawners, oocyte and sperm collection."

Lines 161-162: And for how long was it stored until the next stages occurred? Was any maintenance medium added to the semen for cooling?

Lines 164-165: Why these concentrations of 0.4% sodium chloride solution and 0.8% sodium chloride solution? Are they commercially available? Or were they prepared? If commercial, provide the product name and manufacturer. If prepared, provide the reagent used and the quality of water used for production.

Line 166: Reference [33] does not deal with semen evaluation and there is no information about analysis with the CASA system.

Line 169: This Reviewer has extensive experience with the use of CASA, but has never seen this slide description. It is necessary to inform the model and manufacturer. Is it a product for semen analysis? And 30 μm in depth is too high for analyzing cells as small as those of fish. It should be a maximum of 20 μm in depth. This makes it difficult to focus adequately and count all cells in the field.

Line 174: Verify if CRISMAS was validated for fish and include an image of the software in use.

Line 189: Use "pooled semen" instead of "mixed semen."

Line 196: Describe how the percentage of fertilization and eye formation was assessed.

Lines 241-244: Describe in the methodology how the duration of treatments was assessed.

Line 307: Reference [34] does not support the information about 125 to 235 mOsm/kg.

Line 318: Incorrect reference. Verify and correct.

The CONCLUSIONS section is excessively long and does not comply with the journal's "Instructions for Authors," which requests “Conclusions: This section is mandatory, with one or two paragraphs to conclude the main text.” Therefore, it is necessary to adjust this section, transferring more appropriate parts to the DISCUSSION.

Comments on the Quality of English Language

The technical and scientific use of English is adequate. However, there are some areas where the text could be improved for greater clarity and fluency. In particular, the structure of some sentences is complex and could be simplified, and there is repetition of ideas that could be condensed. Additionally, the use of certain expressions could be adjusted to improve readability and the flow of the text. Thus, adjustments are necessary to enhance the clarity and fluency of the text without compromising the technical content presented.

The reviewer recommends that the text be reviewed by a native English speaker with knowledge of technical terms.

Author Response

  1. Brief Summary

This manuscript investigates the efficacy of different sperm activation and oocyte collection methods on the reproductive outcomes of the pike (Esox lucius). The study demonstrates that the pneumatic method of oocyte collection, combined with a 0.4% NaCl solution, improves fertilization and larval hatching rates. However, the study was conducted nearly a decade ago, and most references are outdated, which may render the manuscript obsolete before publication.

2. General Concept Comments

The manuscript identifies gaps in knowledge regarding sperm activation and oocyte collection in Esox lucius but needs to address the limitations of current methods and consider recent advances, such as commercial media for sperm activation. Why use the NaCl solution (possibly produced rather than a commercial product) when it is well-known that K+ and ions are key factors in controlling spermatozoa motility initiation?

The use of a NaCl solution in the study may stem from traditional approaches or specific experimental conditions that have previously yielded positive results. However, potassium ions (K+) are indeed crucial in regulating sperm motility, particularly in freshwater species such as pike. Fish spermatozoa activate in response to changes in ionic balance, specifically the Na+/K+ ratio, which influences the initiation of motility.

Therefore, it would be beneficial to consider using more complex commercial media for sperm activation, which contain appropriate concentrations of K+ and other ions to better mimic the natural conditions that trigger sperm activation. This could improve reproductive outcomes and enhance the relevance and value of the study.

The opening sentence of the introduction caused significant confusion for this reviewer. When the authors state, "The term controlled reproduction covers all breeding activities carried out to obtain healthy hatching from sexually mature spawners", what exactly do they mean by "controlled reproduction"? Does it cover semen collection and ova insemination? This was unclear in the first paragraph of the introduction.

The term "controlled reproduction" in this context refers to all actions aimed at obtaining offspring under cultured conditions. This includes the collection of sperm from males, the fertilization of eggs, and any other processes that are managed by humans to enhance reproductive efficiency. The process may involve artificial fertilization, as well as the conditions in which broodstock are kept, in order to achieve optimal reproductive outcomes.

For this reviewer, "controlled reproduction" is understood as the management and control of environmental and physiological conditions to stimulate the natural reproduction of fish, where human intervention is minimal. In contrast, "assisted reproduction" refers to more direct intervention techniques to facilitate reproduction, which may include hormonal induction to stimulate ovulation and spermiation, artificial fertilization, and manipulation of gametes (ova and sperm) outside the fish's body. Therefore, there is significant human intervention.

This distinction between "controlled reproduction" and "assisted reproduction" is important as it clarifies the degree of human involvement in the reproductive process. Controlled reproduction aims to create optimal conditions for natural spawning, thereby reducing stress on the fish and promoting their well-being. On the other hand, assisted reproduction employs various techniques that directly influence the reproductive process, which can lead to higher success rates but may also introduce additional stress factors for the fish. By understanding these differences, we can better assess the methodologies employed in aquaculture and their implications for fish health and reproductive success.

The term “assisted fish reproduction” is used by the last author, in Kowalski & Cejko, in the article “Sperm quality in fish: Determinants and affecting factors.” Therefore, it is necessary to clarify why the authors believe “controlled reproduction” is the correct term or to change it to “assisted reproduction.”

In the article by Kowalski and Cejko, where the term "assisted fish reproduction" is used, I believe it would be more appropriate to refer to it as "artificial spawning of pike." The term "artificial spawning" better captures the nature of the process in which humans actively intervene in the reproductive cycle of fish, including their spawning and egg incubation, to enhance reproductive efficiency and production of fry.

In the context of fish farming, particularly for pike, it is crucial to understand the difference between artificial and assisted reproduction. "Artificial spawning" implies full control over all stages of reproduction, which is essential in aquaculture. On the other hand, "assisted reproduction" might suggest that fish are reproducing naturally with minimal assistance, which does not always reflect the reality of artificial spawning.

For this reason, I believe the term "artificial spawning" is more fitting to describe the process presented in the article.

The manuscript lacks a description of the control of experimental variables, such as temperature and pH. Additionally, the sample size is small, compromising statistical significance. The methodology does not clarify the use of sterile gases or the preparation of saline solutions, raising questions about the sterility and safety of the procedures.

The description of sperm activation solutions in the methodology is vague, and the choice of NaCl concentrations is not justified. Reference [33] does not address semen analysis with the CASA system, and the use of slides for analysis requires more details.

The literature review is inadequate, with only two recent references among 69 (45 of which are over ten years old). In a dynamic field like aquaculture, it is essential to update references to reflect the current state of knowledge. Including more recent studies is necessary to contextualize the results.

Due to incorrect numbering of references, this reviewer could not adequately assess the discussion section, as it was impossible to verify the references supporting the discussions. Therefore, this reviewer is unable to evaluate the Discussion section of the present manuscript.

  1. Specific Comments
  • Abstract: The opening sentence creates confusion about what "controlled reproduction" covers. It is recommended to clarify or change to "assisted reproduction."
  • Lines 49-54: Use continuous text instead of bullet points.

Corrected, we used continuous text instead of bullet points.

  • Lines 55-57: Using these lines as an example, this is "assisted reproduction," not "controlled reproduction."

I think you mean artificial reproduction rather than assisted reproduction, and it fits better than assisted reproduction.

  • Lines 69-70: For oxygen or nitrogen, they may be used with medicinal grade—is this the case? How is ambient air sterilized? It seems nonsensical to have a sterile needle if sterile gas isn't used. Could this be clarified? Clarify the use of sterile gases and the sterilization of ambient air. Discuss the choice between oxygen and nitrogen. Is it nitrogen or nitrous oxide? Also, why oxygen or nitrogen? In mammals, CO2 is used as it can be absorbed by tissues and removed through the respiratory system. How does the application of oxygen and nitrogen work?

For pneumatic spawning, only sterile gases are used. The oocytes, along with the introduced air, are freely released from the body cavity of the fish. I also refer you to:

doi:10.3791/56894

doi.org/10.3390/ani10010097

  • Line 85: Old reference. Retain but include new ones. Suggestions:
    https://doi.org/10.1016/j.theriogenology.2023.12.016
    https://doi.org/10.1016/j.anireprosci.2019.01.001
    https://doi.org/10.1016/j.anireprosci.2022.107018

Corrected, we included new ones.

  • Lines 86-88: Not only this. See https://doi.org/10.1016/j.therwi.2024.100091 (interesting considerations about CASA), where it is mentioned that “Evaluation of sperm by conventional microscopy is susceptible to subjectivity, causing significant discrepancies across studies.” If it were only errors of overestimation (and also underestimation), it would be less problematic than the impossibility of replication or replicability of results, which is a cornerstone of the scientific method.

Corrected, we included new one.

  • Lines 116-121: Use continuous text instead of bullet points.

Corrected, we used continuous text instead of bullet points.

  • Lines 117-121: These formulas are from the last century. The newest is 25 years old! In the last two decades, assisted reproduction has evolved considerably, with low-cost, high-efficiency commercial options available. It is important to address this (commercial stable media for sperm activation, such as ActiFish).

Corrected, we included new ones.

  • Line 123: Would it be “fish farmer”? This Reviewer understands "fishing practitioners" refers to people who engage in fishing. This can include professional fishermen, amateurs, or anyone involved in the activity of fishing. What is the objective of a fisherman collecting and activating semen? And being a fish farmer, why not use commercial activation media developed specifically for this purpose, which is not expensive?

Corrected, we replaced "fishing" by "aquacultre".

  • Line 136: When a number starts a sentence, write the number in full. Also, when referring to two or more animals, use plural (males and females). Correct to: "Sixteen females and three males."

Corrected, we changed “16 female and 3 male pike” change to “The 16 female and 3 male pike”

  • Line 138: The use of capitalization in units like mL, L, and μL is guided by conventions in scientific and technical writing to ensure clarity and avoid confusion. The "L" is capitalized to avoid confusion with the number "1" or the lowercase letter "l," which can look similar in certain fonts. Please correct throughout the manuscript.

Corrected, we replaced 'l' with 'L' everywhere.

  • Lines 139-145: The description of the pneumatic method of oocyte collection lacks details on how air pressure is controlled and the possible effects on the oocytes.

The airflow is controlled using a potentiometer. As for the impact on oocytes, this article provides the answer, specifically regarding the effectiveness of pneumatic spawning compared to traditional methods. Additionally, I refer you to the following articles:

doi:10.3791/56894

doi.org/10.3390/ani10010097

  • Line 141: Include an image of the pneumatic method for better understanding.
  • Line 143: Ambient air? Without an air filter to avoid contaminants? As previously questioned, what is the risk to the animal of insufflation with non-sterile air? What is the risk of using non-sterile air and how is this managed?

The air introduced into the fish's body cavity is sterile.

  • Line 146: Use "gametes" instead of "sexual products."

Corrected, we replaced "sexual products" by "gametes".

  • Lines 152-156: This paragraph would be better positioned after “2. Materials and Methods” before the section "2.1. Spawners, oocyte and sperm collection."

I think it’s fine. The description and course of the experiment are presented chronologically.

  • Lines 161-162: And for how long was it stored until the next stages occurred? Was any maintenance medium added to the semen for cooling?

We used the semen immediately after collection. It is pure semen without any cooling or preserving substances.

  • Lines 164-165: Why these concentrations of 0.4% sodium chloride solution and 0.8% sodium chloride solution? Are they commercially available? Or were they prepared? If commercial, provide the product name and manufacturer. If prepared, provide the reagent used and the quality of water used for production.

We studied the semen parameters of pike at varying concentrations up to a 1% sodium chloride solution. We identified two optimal concentrations for the experiment. This is not a commercial product; it was created by us for the purposes of the experiment. The salt solution is a simple two-component solution, and in our opinion, there is no need to describe its production procedure. It consists of salt and water.

  • Line 166: Reference [33] does not deal with semen evaluation and there is no information about analysis with the CASA system.

Corrected, we changed it to the appropriate reference.

  • Line 169: This Reviewer has extensive experience with the use of CASA, but has never seen this slide description. It is necessary to inform the model and manufacturer. Is it a product for semen analysis? And 30 μm in depth is too high for analyzing cells as small as those of fish. It should be a maximum of 20 μm in depth. This makes it difficult to focus adequately and count all cells in the field.

We followed the established procedures regarding pike semen. It is a 12-well slide with a 30-μm depth, Teflon-coated (Tekdon, Inc., Myakka City, FL, USA).

  • Line 174: Verify if CRISMAS was validated for fish and include an image of the software in use.

Yes, it has been approved for use with fish. For confirmation, there are articles where the CRISMAS software has been used for fish semen analysis. For example:

https://doi.org/10.1016/j.theriogenology.2019.10.035
https://doi.org/10.1016/j.aquaculture.2019.734482 https://doi.org/10.1016/j.aquaculture.2016.06.015 https://doi.org/10.1016/j.anireprosci.2018.04.073

  • Line 189: Use "pooled semen" instead of "mixed semen."

Corrected, we replaced "mixed semen" by "pooled semen".

  • Line 196: Describe how the percentage of fertilization and eye formation was assessed.

The fertilized eggs were placed in tanks, 100 per tank. Based on this, the percentage of fertilized oocytes was calculated, followed by the hatching rate.

  • Lines 241-244: Describe in the methodology how the duration of treatments was assessed.

Assessed using a stopwatch.

  • Line 307: Reference [34] does not support the information about 125 to 235 mOsm/kg.

Corrected, we changed it to the appropriate reference.

  • Line 318: Incorrect reference. Verify and correct.

Corrected, we changed it to the appropriate reference.

The CONCLUSIONS section is excessively long and does not comply with the journal's "Instructions for Authors," which requests “Conclusions: This section is mandatory, with one or two paragraphs to conclude the main text.” Therefore, it is necessary to adjust this section, transferring more appropriate parts to the DISCUSSION.

Comments on the Quality of English Language

The technical and scientific use of English is adequate. However, there are some areas where the text could be improved for greater clarity and fluency. In particular, the structure of some sentences is complex and could be simplified, and there is repetition of ideas that could be condensed. Additionally, the use of certain expressions could be adjusted to improve readability and the flow of the text. Thus, adjustments are necessary to enhance the clarity and fluency of the text without compromising the technical content presented.

The reviewer recommends that the text be reviewed by a native English speaker with knowledge of technical terms.